# Evaluation of the One Health-Ness of 20 Years of Antimicrobial Resistance Surveillance in Norway

**DOI:** 10.3390/antibiotics12071080

**Published:** 2023-06-21

**Authors:** Madelaine Norström, Gunnar Skov Simonsen, Jannice Schau Slettemeås, Anne-Sofie Furberg, Anne Margrete Urdahl

**Affiliations:** 1Norwegian Veterinary Institute, N-1431 Ås, Norway; 2Department of Microbiology and Infection Control, University Hospital of North Norway, N-9038 Tromsø, Norway; 3Faculty of Health Sciences, UiT-The Arctic University of Norway, N-9038 Tromsø, Norway

**Keywords:** One Health, antimicrobial resistance, surveillance, evaluation, OH-EpiCap

## Abstract

We evaluated the One Health-ness (OH-ness) of the surveillance system for antimicrobial resistance (AMR) in Norway by using the recently developed “Evaluation tool for One Health epidemiological surveillance capacities and capabilities” (OH–EpiCap tool). First, we defined the Norwegian AMR surveillance system that we would evaluate. The tool was applied by a group of stakeholders (key persons in the Norwegian AMR surveillance programmes and authors of this paper). The evaluation was performed using a consensus approach. The evaluation resulted in an overall OH-ness score of 68% across all three dimensions included in the tool: Organisation, Operation, and Impact. Suggestions for improvement were only indicated within the areas of internal evaluation and operational costs, whereas most of the indicators included in the tool showed good adherence to the One Health principles. By performing this internal evaluation, we recognized that AMR surveillance in the environment needs to be included in a more systematic and standardized way to improve the OH-ness as defined by the quadripartite organisations. Last but not least, it was beneficial to bring key stakeholders together to conduct the evaluation. It increased a joint perception of the OH-ness of AMR surveillance in Norway and encouraged further collaboration in the future.

## 1. Introduction

The Quadripartite Partners, i.e., the Food and Agriculture Organisations of the United Nations (FAO), the World Organisation for Animal Health (WOAH), the World Health Organisation (WHO), and the United Nations Environment Programme (UNEP), promote a One Health (OH) integrative approach for the monitoring of antimicrobial usage and resistance [1]. Recently, the OH integrative approach was proposed by Aenishaenslin et al. [2] as “the systematic collection, validation, analysis, interpretation of data, and dissemination of information collected on humans, animals, and the environment to inform decisions for more effective evidence-and system-based health interventions”. The OH approach recognizes that the emergence and spread of antimicrobial resistance (AMR) is a complex and multifaceted issue that requires a holistic view of the various interconnected factors that contribute to it. Such surveillance systems require collaboration and coordination among several stakeholders, including public health agencies, veterinary organisations, and environmental agencies. These efforts involve sharing data, expertise, and resources, as well as developing common strategies for surveillance and control.

Norway was one of the European countries that took the initiative to monitor AMR in both the veterinary sector and the human sector in the late 1990s. The Norwegian monitoring programme for AMR in the veterinary sector (NORM-VET) started in 1999, and the Norwegian surveillance programme for AMR in human medicine (NORM) was initiated in the following year. The programmes have been collaborating since 2000, publishing a joint annual report, NORM/NORM-VET Usage of Antimicrobial Agents and Occurrence of Antimicrobial Resistance in Norway [3,4]. It should be noted that although usage data for both the human and veterinary sectors are presented in the report, they are not formally included in the NORM and NORM-VET programmes. These programmes have been developed side by side, recognizing the different opportunities and limitations for monitoring the antimicrobial susceptibility of microbes in the human and veterinary sectors. In addition to the monitoring and surveys included in NORM-VET, a yearly surveillance programme of livestock-associated methicillin-resistant *Staphylococcus aureus* (LA-MRSA) in swine has been running since 2014. AMR surveillance in NORM consists of data from reference laboratories for various microorganisms, mandatory reporting of specific resistance phenotypes through the Norwegian Surveillance System for Communicable Diseases (MSIS), as well as routine data from clinical laboratories. Mapping of AMR in different environmental niches has also been conducted for the Norwegian Environment Agency since 2016.

To date, an evaluation of the OH-ness of the AMR surveillance programmes in Norway has not been performed. Performing such an evaluation is, however, recommended [2,5]. There are several available tools, some of which have been assessed in previous studies [6,7,8]. The NORM-VET programme was used as a case study in one of these comparison studies [7]. However, the study focused on the evaluation of some specific tools and not the surveillance programmes per se. Overall, the selection of an evaluation tool should depend on the purpose of the evaluation.

Recently, the “Matrix” Consortia developed a new tool, called the “Evaluation tool for One Health epidemiological surveillance capacities and capabilities” (OH-EpiCap), with support from the One Health European Joint Programme Initiative (OH-JPI) [9,10]. The OH-EpiCap tool aims to identify and describe collaborations among actors and sectors involved in the surveillance of a pathogen/hazard and to characterize the OH-ness of the surveillance systems by the use of a set of indicators. We considered the OH EpiCap tool to fit our need for an internal evaluation of the OH-ness of AMR surveillance in Norway, with the aim of using the evaluation to improve the current surveillance system.

The evaluation was initiated within the JPI network, “Convergence in evaluation frameworks for integrated surveillance of AMR, (CoEval-AMR PHASE2)”, when the consortium had the opportunity to use a beta version of this tool, both for assessment of the tool [11,12] and to perform a full evaluation of the OH-ness of the AMR in several case studies. In the present study, we report on the evaluation of the surveillance of AMR in Norway.

## 2. Results

The evaluation resulted in an overall score of 68% across all three dimensions: Impact, Organisation, and Operation (Figure 1). The scores within each dimension were highest for the Impact, at 79% (Figure 2), followed by the Organisation, at 67% (Figure 3), and the Operation, at 57% (Figure 4). These results are also included in the Appendix A, which is the report generated by the OH-EpiCap tool itself, in which the comments for each indicator can be found.

Table 1 shows the indicators demonstrating good adherence to the OH principles and the indicators that would benefit the most from improvement, according to the evaluation.

## 3. Discussion

Here we present an evaluation of AMR surveillance in Norway using the OH-EpiCap tool. To our knowledge, this is the first evaluation of the “OH-ness” of AMR surveillance in Norway. The results from the evaluation identified only a few areas that could be improved, namely the internal evaluation and the operational costs.

The operational costs of surveillance in the veterinary sector are, to some extent, not under the influence of NORM-VET per se, given that it is primarily designed to fulfil the requirements set by the European Union (EU)/European Food Safety Authorities (EFSA) and/or the Norwegian Government’s National Strategy against AMR, with the subsequent sector-specific AMR Action Plan from the Norwegian Ministry of Agriculture and Foods. Sector-specific surveillance activities may have goals of importance that overrule any considerations of operational costs. For NORM, the surveillance is based on collecting already-existing data, and operational costs are thereby limited. Therefore, we did not consider the operational cost indicator relevant in our evaluation. As for the indicator of internal evaluation, we address this in the present study by performing an internal evaluation with key stakeholders from the human and veterinary sectors (i.e., animals and food). It should be noted that the veterinary sector has also been involved in the majority of the survey activities performed in the environmental sector.

The evaluation resulted in an overall OH score of 68% across the three dimensions of Organisation, Operation, and Impact. In the following sections, we will discuss our results, within each dimension, for the achieved scores for each of the indicators related to the targets included.

### 3.1. Organisation

According to the Norwegian Government’s National Strategy against AMR, there is an overarching cross-sector aim to keep the prevalence of AMR in Norway as low as possible [13]. A top score of four (4) was therefore set for a common aim in OH surveillance, although regular surveillance of the environment has not been established. In addition, both NORM and NORM-VET are designed to monitor and detect any changes in AMR occurrence.

The other questions that related to the target “formalization” had lower scores, given that supporting documentation (score: 2), leadership (score: 3), and coordination (score: 3) are not shared between the sectors, as this is not considered necessary or relevant under normal conditions. In the case of an emergency, however, coordination will rapidly be established between all relevant sectors based on national emergency plans.

The surveillance in the human and veterinary sectors is designed to be representative of the populations of concern, as well as the geographic area of Norway. However, as mentioned, the environmental sector is not systematically included in any regular surveillance. This has an impact on the scores given for all questions related to coverage and transdisciplinarity (i.e., scores of three (3)).

Budget and human resources are in place; however, there is no specific budget allocated to fulfil all the designated surveillance targets from the Norwegian Ministry of Agriculture and Food’s sector-specific AMR Action Plan. This includes, for instance, AMR surveillance of family and sports animals, and food and feed, which are not included in the EU/EFSA regulations. Nor is a specific national budget allocated for surveillance within the environmental sector, though some money has been allocated for more limited surveys.

External evaluations have been performed by both the European Centre for Disease Prevention and Control (ECDC) and the EFSA on how well Norway is following the international requirements, and most of the corrective measures have been implemented based on these recommendations. To address the occurrence of critical situations, we considered that the systems have enough adaptability to change within an appropriate timeframe.

### 3.2. Operation

The operation score was the lowest at 57%. Although no specific indicator could be detected as the primary cause of this score, it may be explained by the fact that only one of the indicators related to the target communication, namely “Dissemination”, received the score of four. The other indicators received scores of three or two, as these indicators were found to be less integrated. The human and veterinary sectors have different approaches when it comes to protocol design and data collection (i.e., sampling strategies, sampling sizes). There are both differences and similarities among the laboratory techniques used, in particular for susceptibility testing of bacterial isolates, as described in Section 4.2. The statistical analyses are performed in each system and are considered fit for their purpose within each sector. Data sharing, such as data quality, usefulness, and FAIR principles, are handled within each sector separately, and the same applies to data analyses and interpretations. A detailed data-sharing agreement is not deemed necessary or relevant, as information and bacterial strains are only exchanged in specific situations under the general legal framework for human and animal health. Integrative analyses have not routinely been part of the programmes, but some comparative studies have been performed using data from NORM and NORM-VET in different research projects. Examples of such studies are *C. jejuni* isolates from broilers and humans infected abroad vs. in Norway [14] and cephalosporin-resistant *E. coli* from chicken meat and Norwegian patients [15]. The internal communication between the actors is informal and flexible but could probably be improved. However, in the case of an emergency, information will be shared between all relevant actors and sectors within a short period. External communication and dissemination are jointly performed, especially concerning the publication of the annual report, and received the highest score of four. However, the environmental sector is not included in these activities as this sector lacks regular systematic surveillance. Separate reports have been published for the environmental surveys, although data from some have also been presented in the NORM/NORM-VET reports.

### 3.3. Impact

The indicators belonging to the target technical outputs—emergence detection, improved knowledge, effectiveness, and operational costs—showed the largest variations in scores.

AMR emergencies are detected within relatively short timeframes, though any reduction in time of detection, would be of advantage to ensure a better management of the situation. Real-time detection is rarely relevant, and rapid alert systems are in place for the most important hazards.

Operational costs have never been evaluated, and we assessed this not to be relevant for AMR surveillance in Norway. In the human sector, surveillance is based on existing data, and in the veterinary sector, the majority of the activities performed are regulated by the EU/EFSA/ECDC and the national authorities and, thereby, are unavoidable costs. However, a socio-economic OH evaluation was performed to assess different surveillance options and strategies before the implementation of the surveillance programme of LA-MRSA in swine [16,17].

The collaborative added value of the OH system included indicators concerning the OH team, the OH network, international activities, and strategy, which all had scores between three (3) and four (4). The following comments were given, such as:

“The OH team is informal, and there are not that many actors involved, therefore, the persons involved know each other quite well. This facilitates good collaboration and the production of the joint report. The fact that we do write a joint report also has an impact on maintaining the OH collaboration”.

“An updated National AMR Strategy is currently under development by national authorities. AMR surveillance data have been important in previous strategy development, and the OH aspects will presumably be a defining topic in the new version as well”.

“Concerning international activities, actors from both human and veterinary AMR surveillance in Norway are actively involved in different initiatives within ECDC, EFSA, WHO, FAO and OIE forums”.

All the indicators of the targets immediate and intermediate outcomes—preparedness, interventions and advocacy—adhered to the OH-ness of the Impact dimension. We found that the surveillance systems are well suited to detect and respond to any relevant AMR emergencies. An example of such a response was the rapid implementation of MRSA screening of foreign workers at pig farms and of human patients at risk before hospital admissions. Regarding preparedness, there is a concern that resistant clinical isolates from animals can go undetected, as a large proportion of samples from sick animals are sent to laboratories outside Norway and therefore not included in the national surveillance. Examples of interventions that have been implemented as a result of data from the surveillance programmes are the actions taken to keep the levels of MRSA- and ESBL-producing bacteria at the lowest possible levels. Here, the collaboration between the human and veterinary sectors has led to a policy change for LA-MRSA in the swine population [17]. For *E. coli* with ESBLs (mainly AmpC) in broiler production, interventions were implemented by the poultry industry itself, and, as a result, these bacteria are now almost absent in Norwegian broiler production [18]. Similarly, the use of the coccidiostat narasin has been phased out in the Norwegian broiler industry since 2016 [19,20]. Several stakeholders found the use of such agents controversial due to their additional antibacterial effect.

The results from the annual reports are disseminated through several media channels and joint seminars, including relevant stakeholders. Although there are no formal studies of the impacts, these activities have certainly led to increased awareness of AMR and have also improved the knowledge regarding AMR among the stakeholders. Chronicles, seminars, interviews, and informational materials to raise awareness in the public and among stakeholders are examples of the extensive advocacy activities that have been performed.

As one of the ultimate outcomes, the existing OH surveillance system has initiated and facilitated several multisectoral research collaborations. A few examples of such projects are QREC-risk (NFR: 244140), QREC-MaP (NFR: 25016), and NoResist (NFR: 250212).

We have mentioned policy changes regarding LA-MRSA in the previous section, and some specific resistance phenotypes have also been made notifiable when detected in animals [21]. In the human sector, new guidelines for infection control and antimicrobial prescribing have been implemented. Behavioural changes can be seen, such as the phasing out of anticoccidials in the poultry sector, the decreased amount off antimicrobials used for animals, and the wide use of facemasks to prevent the spread of LA-MRSA. Last but not least, the ultimate indicator belonging to the Impacts category was the health outcome. This was quite difficult to assess, particularly for Norway, as the occurrence of AMR is very low, and the surveillance activities performed have the objective of maintaining the current situation. A negative effect on the health outcome might have been the case if no surveillance had been performed at all, but this is speculative, and our score on this indicator was therefore only two (2). At present, there is not any consensus on how to monitor AMR in the environment. This was also stated in a recent Scientific Opinion from the Norwegian Scientific Committee for Food and Environment [22]. They observed that calls for surveillance are not uniform nor straightforward to implement, that the rationale for surveillance in the environment differs from that in humans, and that tools, techniques and determinates need to be standardised and harmonised at an international level. Still, the Committee concluded that there is a scientific rationale, methodological opportunity, and broad support for the establishment of an environmental AMR surveillance programme in Norway and that such surveillance should focus on the effects of anthropogenic practices on the environment.

During the evaluation, the tool guided us through the questions and encouraged a discussion among the participants. We found this to be a valuable exercise, which gave us the opportunity to reflect on the relevance of the questions and the reasons for the resulting scores. Our general impression was that, although the OH-EpiCap is designed with a score of four (4) as the desired outcome, the optimal score will depend on the aim and organisation of the surveillance programme under evaluation. Furthermore, the OH-EpiCap tool is not specifically designed for the evaluation of AMR surveillance, and it only includes three dimensions. To the authors’ knowledge, there are currently no evaluation tools available that include all possible dimensions. Thus, the choice of tool used in any evaluation will depend on the study’s aim and available resources. Concerning the current evaluation using the OH-EpiCap tool, we found the tool easy to use and that the generated outputs were valuable for further discussions. A thorough evaluation of the OH-EpiCap tool has recently been performed within the JPI CoEval-AMR network [12].

## 4. Materials and Methods

### 4.1. Definition of the One Health System of AMR Surveillance

In this paper, we define the OH system of AMR surveillance in Norway to include the Norwegian AMR monitoring programmes in the human (NORM) and veterinary sectors (NORM-VET), the surveillance programme for LA-MRSA in swine, parts of the MSIS, as well as the AMR mapping surveys performed in different niches of the Norwegian environment. The term “sectors” refers to the different sectors, namely veterinary (animal, feed, and food), human, and environment, and the term “actors” refers to the persons involved in the different sectors.

### 4.2. Description of the Different Sectors of the One Health System of AMR Surveillance

In brief, NORM is based on periodic surveys of specific microorganisms from various clinical specimen types and uses standardised bacteriological methodologies. All included bacterial isolates are stored for further analyses if needed. The protocol for NORM is revised yearly, but several key parameters (microorganisms, antibiotics, specimen types) remain unchanged in order to facilitate comparisons over time. All clinical microbiology laboratories in the country participate, and the quality, integrity, and completeness of data are ensured through a custom-made IT platform (eNORM). Susceptibility testing is mainly done by disk diffusion or MIC gradient strips. The results are reported as continuous variables, but, for publication purposes, they are most often categorized according to EUCAST breakpoints as susceptible with standard exposure (S), susceptible with increased exposure (I), or resistant (R). All published materials, including historical data, are based on the most recent version of the breakpoint protocol. The national MSIS register includes some specific notifiable infections and carrier ships caused by resistant microbes: MRSA, penicillin-resistant pneumococci, vancomycin-resistant enterococci, and selected resistance patterns among Enterobacterales, *Pseudomonas* spp., and *Acinetobacter* spp. (i.e., colistin resistance and carbapenemase production). Reference laboratories for specific organisms are tasked to monitor antimicrobial susceptibility patterns in their respective reference organisms.

The data recorded for susceptibility testing in NORM-VET are mainly derived from healthy animals using the indicator bacteria *Escherichia coli* and *Enterococcus* spp. (i.e., *E. faecalis* and/or *E. faecium*) from faecal or caecal samples. Zoonotic bacteria such as *Campylobacter* spp. (i.e., *C. coli* and/or *C. jejuni*) and *Salmonella* spp. are also included. The *Salmonella* isolates originate from the Norwegian surveillance programme for *Salmonella* in broilers, swine, and cattle [23], as well as from diagnostic submissions. In addition, clinical bacterial isolates from sick animals are included, although they are limited to bacteria where a sufficient number of isolates, from one or several years, are available. Susceptibility testing is mainly performed using microtiter plates and recorded as minimum inhibitory concentration values (MICs). The results are reported as MIC distributions and categorised as resistant or susceptible based on epidemiological cut-off values (ECOFFSs), mainly based on recommendations from EUCAST. The NORM and NORM-VET programmes operate separately, but there are similar laboratory techniques for specific resistant bacteria, such as MRSA and extended-spectrum beta-lactamase (ESBL)-producing bacteria. The NORM and NORM-VET programmes have undergone many adaptations over the years. In particular, the NORM-VET programme has been adjusted to follow the requirements set by the EU/EFSA to harmonise sampling, susceptibility testing, and reporting to the EU. This has impacted sample sizes, susceptibility test panels, and the cut-off values [24,25]. After these changes were implemented in 2014, the objective and focus on the OH aspects were greater than in previous years. Antimicrobial substances that are important for human medicine are now included in the panels, and the sampling of food (i.e., meat) is risk-based, including proportionate sampling at retailers, depending on the size of the human population in the regions of each country. The surveillance programme for LA-MRSA in swine includes the census population of all swine herds in Norway and has a search and destroy policy [26]. The AMR surveys performed in different niches of the environment have thus far been independent and not regulated or put into a system.

### 4.3. Methodology

A core group of key persons who are involved in the NORM and NORM-VET programmes applied the generic benchmarking tool OH-EpiCap. The OH-EpiCap tool has recently been developed (and is still under development) and consists of a semi-quantitative tool that is designed to assess the OH-ness of a surveillance system. It is highly recommended to use the tool within a workshop setting that includes a panel of representatives across the sectors.

Here, the core group consisted of the editors and main contributors to the yearly AMR surveillance report NORM/NORM-VET, from both the human and veterinary sectors, who have more than 20 years of experience working within AMR surveillance in Norway. Together, this core group covers the design and management of the programmes, laboratory methods, data analysis and data curation, dissemination, and communication with relevant stakeholders, including the government, industry, hospitals, and the public.

A beta version of the OH-EpiCap tool was accessed at https://carlijnbogaardt.shinyapps.io/OH-EpiCap on 15 August 2022. The tool was used several times by the first author before the current evaluation was performed. This was to get acquainted with the tool before performing an evaluation. The questions in the tool were answered and discussed in a digital meeting with all the authors (i.e., key persons) on 6 February 2023. A consensus was agreed on, and only one score was given for each question. The tool produced a report on the OH-EpiCap website, with dimension indices representing mean scores for all questions, expressed as percentages. A newer version of the tool can be found at https://freddietafreeth.shinyapps.io/OH-EpiCap/, accessed on 25 April 2023. A link to the user guide is provided within the tool itself.

The OH-EpiCap tool includes three so-called dimensions: 1. Organisation, 2. Operation, and 3. Impact. Four targets have been defined for each of the three dimensions, as shown in Table 2. The four target areas included under Organisation are, namely, formalization, coverage, resources, and evaluation and resilience. Targets under Operation include data collection and methods sharing, data sharing, data analysis and interpretation, and communication. Lastly, technical outputs, collaborative added value, immediate and intermediate outcomes, and ultimate outcomes are included under the Impact dimension. In total, there were 48 questions, with four questions per target. The obtained answers resulted in a score ranging from one (1, no compliance) to four (4, full compliance). An option of not applicable (NA) was available if the question was considered not to be relevant. There was also an option to reformulate the questions to adapt them to the surveillance system under evaluation. At present, the dimension indices represent mean scores over all questions, expressed as percentages.

## 5. Conclusions

Overall, we found that the OH-ness of AMR surveillance in Norway, as evaluated by the OH-EpiCap tool, was rather good. However, including systematic surveillance of the environment would help to improve the OH-ness of the overall AMR surveillance system. Last but not least, it was beneficial to bring key stakeholders together to conduct the evaluation. It increased a joint perception of the OH-ness of AMR surveillance in Norway and encouraged further collaboration in the future.

## Figures and Tables

**Figure 1 antibiotics-12-01080-f001:**
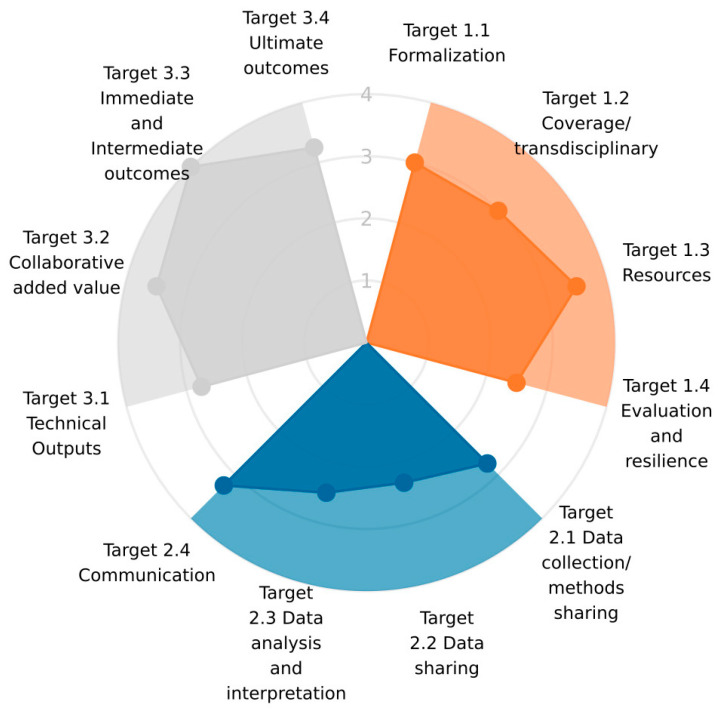
The average scores of the target areas attributed to three dimensions, Organisation (in orange), Operation (in blue), and Impact (in grey), from the evaluation performed on the surveillance of antimicrobial resistance in Norway using the OH-EpiCap tool.

**Figure 2 antibiotics-12-01080-f002:**
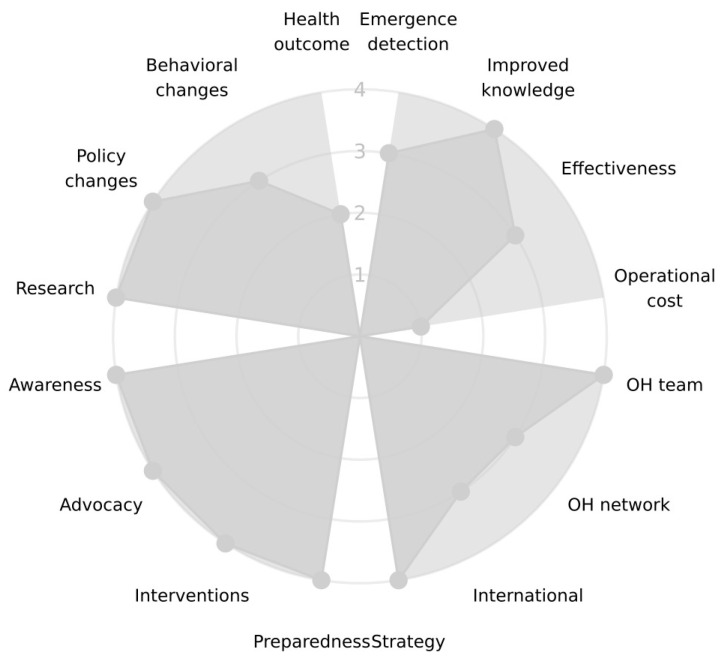
The scores (1–4) of the indicators within the four targets of the Impact dimension, from the evaluation performed on the surveillance of antimicrobial resistance in Norway using the OH-EpiCap tool.

**Figure 3 antibiotics-12-01080-f003:**
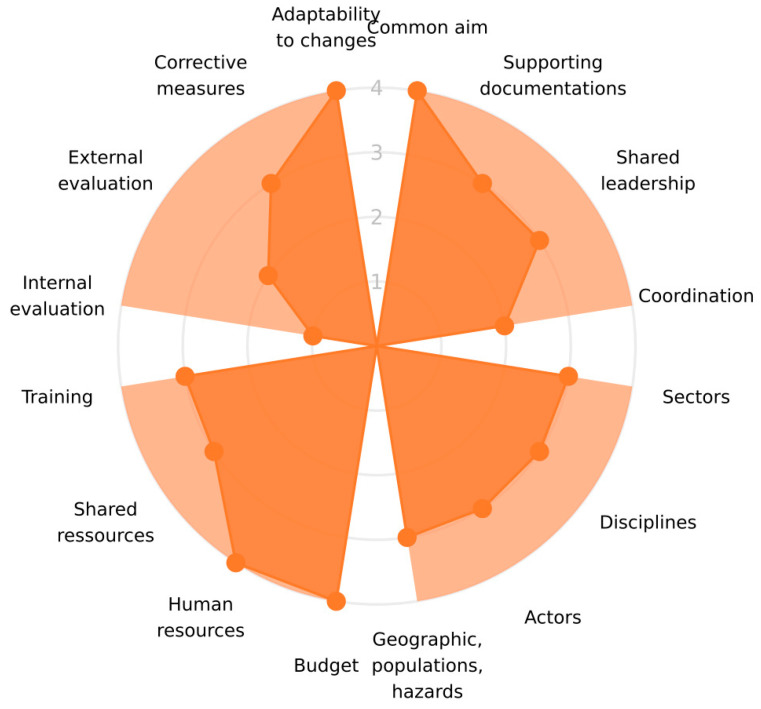
The scores (1–4) of the indicators within the four targets of the Organisation dimension, from the evaluation performed on the surveillance of antimicrobial resistance in Norway using the OH-EpiCap tool.

**Figure 4 antibiotics-12-01080-f004:**
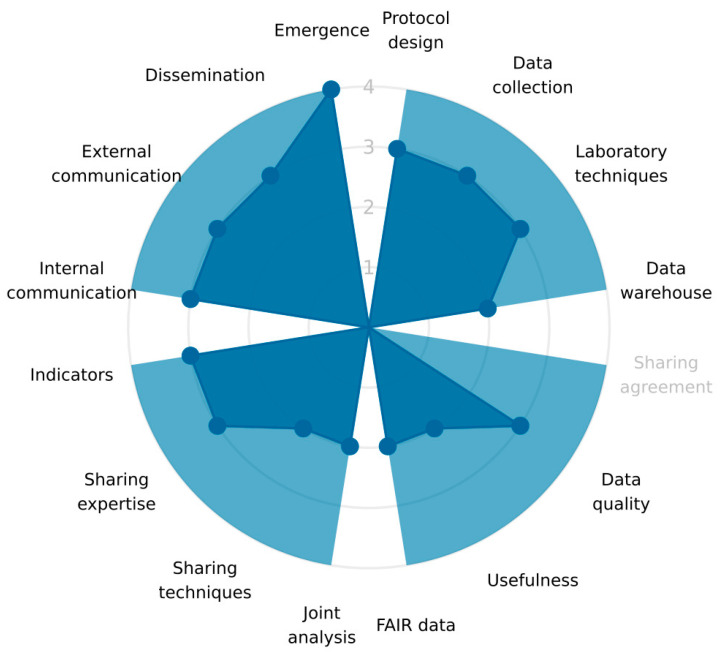
The scores (1–4) of the indicators within the four targets of the Operation dimension, from the evaluation performed on the surveillance of antimicrobial resistance in Norway using the OH-EpiCap tool.

**Table 1 antibiotics-12-01080-t001:** Indicators within the three dimensions, Organisation, Operation, and Impact, that were identified by the OH-EpiCap tool to have good adherence to the OH principles and the ones that could be improved according to the OH principles.

	Dimensions		
Description	Organisation	Operation	Impact
Indicators showing good adherence to the OH principles	common aim budget human resources adaptability to changes	emergence	improved knowledge OH team strategy preparedness interventions advocacy awareness research policy changes
Indicators that could be improved according to the One Health principles	internal evaluation	none	operational costs

**Table 2 antibiotics-12-01080-t002:** An overview of the dimensions, targets, and questions that are covered within the targets and dimensions covered by the OH-EpiCap tool for assessing the OH-ness of a surveillance system as described in the tool https://freddietafreeth.shinyapps.io/OH-EpiCap (accessed on 26 April 2023). Each of the targets included four questions, which could be scored from one (1, no compliance) to four (4, full compliance). The table has been adapted using the same terminology as in the tool itself.

Dimensions	Targets	Questions About:
Organisation	formalization	“the objectives, supporting documentation, coordination roles, leadership”
	coverage	“whether all relevant actors and disciplines, sectors geography, populations and hazards are covered”
	resources	“availability of financial and human resources, training and sharing of the available operational resources”.
	evaluation and resilience	“internal and external evaluation, implementation of corrective measures and the capacity to adapt to changes”
Operations	data collection and methods of sharing	“multisectoral collaborations in the design of surveillance protocols, data collection, harmonisation of laboratory techniques and data warehousing”
	data sharing	“data sharing agreements, evaluation of data quality, use of shared data, compliance with the FAIR principle”.
	data analysis and interpretation	“multisectoral integration of data analysis, sharing of statistical techniques and scientific expertise, harmonization of indicators”
	communication	“internal and external communication processes, dissemination to decision-makers, information sharing in case of suspicion”
Impact	technical outputs	“Timely detection of emergence, knowledge improvement on hazard epidemiological situations, increased effectiveness of surveillance, reduction of operational costs”
	collaborative added value	“strengthenings of the OH-team and network, international collaboration and common strategy”
	immediate and intermediate outcomes	“advocacy, awareness, preparedness, interventions based on the information generated by the OH-surveillance system”
	ultimate outcomes	“research opportunities, policy changes, behavioural changes, better health outcomes that are attributed to the OH-surveillance system”

## Data Availability

Data is contained within the article or Appendix A.

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
