# Peer review of "Evaluation of the One Health-Ness of 20 Years of Antimicrobial Resistance Surveillance in Norway"

_antibiotics, 2023, doi:10.3390/antibiotics12071080_

Round 1

Reviewer 1 Report

The authors describe the use of OH-Epi-Cap tool for evaluation of the surveillance of AMR in Norway. However, an extensive analysis of AMR within the three dimensions Organisation, Operation and Impact is essential for a better understanding.

The quality of english language is fine

Author Response

Comments and Suggestions for Authors

The authors describe the use of OH-Epi-Cap tool for evaluation of the surveillance of AMR in Norway. However, an extensive analysis of AMR within the three dimensions Organisation, Operation and Impact is essential for a better understanding.

Answer:

The paper describes the use of the OH-EpiCap tool and the answers as well as the comments and arguments for the scores that were given when applying the tool. It was not under the scope of the paper to elaborate more on tool itself, however the tool itself is described in reference 6 (https://doi.org/10.1101/2023.01.04.23284159) and also on the OH-EpiCap Tool webpage as shown under the Methodology chapter.

Reviewer 2 Report

Dear Authors,

The presented manuscript is solely a report of a network meeting. It presents the results of conversations etc.

I do not recommend to accept this manuscript, due to a fact that this is neither a case study, nor a research paper, nor a review paper.

Please consider publishing it as a report, without submitting it to an IF journal.

Author Response

Dear Reviwer,

The editor seems to diagree with you that it's not suitable for the journal.

We have understand that this is a special number of Antibiotics where this paper fits well, otherwise we would have submit this paper to another journal as you are quite right.

Reviewer 3 Report

I have evaluated the manuscript (Antibiotics-2346997) titled “Evaluation of the One Health-ness of 20 years surveillance of antimicrobial resistance in Norway” by Norström and coworkers, and the author in this manuscript evaluated the One Health-ness (OH-ness) of the surveillance systems for antimicrobial resistance (AMR) i.e monitoring of antimicrobial usage and resistance, in Norway using developed OH–EpiCap tool. I found this article interesting for the readers and followed the journal Antibiotics’ scope. In the manuscript all methods are explained in detail and results provide valuable insight of the one health-ness system. Overall presentation and discussion of this manuscript with diagram would generate interest from the readers.

I would recommend this article be published in Antibiotics after minor corrections. 

The author needs to address the following comments/corrections.

 1.     The author should correct the format of references wherever needed (e.g Year Bold, Volume Italic etc.).

2.     The author needs to improve the resolution of figure 1-4.

3.     Name of Table 2 should be Table 1, as it appears first in the manuscript, and same applied to Table 1.

4.     The author could have discussed a bit about OH-Epi-Cap tool.

5.     The author could have included the criteria for the scoring system used in OH-EpiCap is from one (1) to four (4) in the tabular form.

6.     A conclusion with expert comments is missing.

7.     In line 207, is it “actors” or sectors, please check this error throughout the manuscript.

8.     The author should make a table for inclusion and exclusion criteria.

No comment.

Author Response

Answer to 3rd Reviewer 

  1. The author should correct the format of references wherever needed (e.g Year Bold, Volume Italic).

Answer: This has been done, accordingly.

  1. The author needs to improve the resolution of figure 1-4.

Answer:  New figures with better resolution are now included, as well as an additional acknowledgement statement to the OH-EpiCap tool developers for their help.

  1. Name of Table 2 should be Table 1, as it appears first in the manuscript, and same applied to Table 1.

Answer: This is now changed accordingly in the text.

  1. The author could have discussed a bit about OH-Epi-Cap tool.

Answer: A thorough evaluation of the OH-EpiCap tool was out of scope for the current paper, though we do agree that it could be more briefly discussed. However, another recently submitted manuscript (of which the first author is co-author) have evaluated and thoroughly discussed the tool. Thereby we decided not to include this part in the present study. The following part is attended at line 128-135: “Further, the OH-EpiCap tool is not specifically designed for evaluation of AMR surveillance, and it only includes the three mentioned dimensions. To the authors’ knowledge, there are at present no evaluation tool available including all possible dimensions   The choice of tool will in any evaluation cases depend on the study aim and available resources. With respect to the current evaluation using the OH-EpiCap tool, we found the tool easy to use and that the outputs generated were valuable for further discussions. A thorough evaluation of the OH-EpiCap tool has been performed within the JPI_CoevalAMR network (submitted).”

  1. The author could have included the criteria for the scoring system used in OH-EpiCap is from one (1) to four (4) in the tabular form.

Answer: As it is four questions for each of the targets, we rather included a sentence in the table text about the scoring system.  Each of the target included four questions which could be scored from one (1) to four (4) as well as included a reference [9] to Tegegne et al. 2023;_ lines 345-348

  1. A conclusion with expert comments is missing.

Answer: Yes, this was not required according to the journal format. However, we have now included a conclusion in lines 316-319:  The OH-ness of the AMR surveillance in Norway, as evaluated by the OH-EpiCap tool, is rather good. Further improvement of the OH-ness should include a systematic environmental AMR surveillance.

  1. In line 207, is it “actors” or sectors, please check this error throughout the manuscript.

Answer: In line 207, now 231 it is actors, the persons involved in the different sectors. Sectors means the different sectors; veterinary (animal, feed and food), human and environment. In that sense we have went through the manuscript to check if the terms have been used wrongly or not. In line 130 and 236 we changed to sector as this seem more appropriate The tool provides a glossary where actor is defined as “An individual or organization that operates with a primary intent to improve health of people, animals and the environment”

  1. The author should make a table for inclusion and exclusion criteria.

Answer: This evaluation was performed on the official AMR surveillance programmes in Norway as described in the Material and methods. There was no other inclusion or exclusion criteria for the AMR surveillance system to be evaluated. The evaluation was performed using the OH-EpiCap tool, answering all questions included in the tool .All scores and answers from the evaluation are then reported and discussed further. The questions included in the tool can be found at the tool webpage.

Reviewer 4 Report

Comments for authors

  1. Title: There seem to be so much use of the word ‘’of” in the title.  Consider the suggested title below

“Evaluation of the One Health-ness of 20 Years Antimicrobial Resistance Surveillance in Norway”

2.      Line 13: What is “OH–EpiCap tool”? It’s important to explain the meaning of this acronym at this first use.

3.      Lines 13-16: Whose consensus please? That of the authors or the “stakeholders”?

4.      Conclusion in the abstract: How much of OH-ness is the 68% overall score of AMR surveillance found in Norway?

5.      Lines 28-32: Citation(s) required

6.      Lines 35-42: Citations required to validate various categorical statements and claims made

7.      The meaning, concept, relevance and applicability of “OH–EpiCap tool” in AMR surveillance in Norway should be well explained in the introduction

8.       Figure legends 1-4: remove periods (.) after the last word in the legend.

9.      Tables: Where is Table 1? Tables should be cited chronologically i.e. Table 1 before Table 2.  Again, remove the periods (..) after the Table 2 title.

10.  Lines 96-97: The first evaluation of OH-ness of AMR surveillance in Norway or the first evaluation of OH-ness of AMR surveillance in Norway using the OH-EpiCap tool?

11.  Lines 131-134: This should be mentioned as a limitation of this study

12.  Discussion: The OH-ness of the 68% overall AMR surveillance is largely not discussed.  Again, the authors need to discuss the significance of low 57% OH-ness obtained for the operationalization component of the triad and suggest how the enhance operationalization with the OH-EpiCap tool.

Author Response

Answer to 4th reviewer

  1. Title: There seem to be so much use of the word ‘’of” in the title.  Consider the suggested title below

“Evaluation of the One Health-ness of 20 Years Antimicrobial Resistance Surveillance in Norway”

Answer: We appreciate this suggestion and have changed the title as suggested!

  1. Line 13: What is “OH–EpiCap tool”? It’s important to explain the meaning of this acronym at this first use.

Answer: We have included at Line 13: developed “Evaluation tool for One Health epidemiological surveillance capacities and capabilities (OH–EpiCap tool).

  1. Lines 13-16: Whose consensus please? That of the authors or the “stakeholders”?

Answer: Clarification at Lines 16 -17; stakeholders (key persons in the Norwegian AMR surveillance programmes and authors to this paper)

  1. Conclusion in the abstract: How much of OH-ness is the 68% overall score of AMR surveillance found in Norway?

Answer: This is a total average score of OH-ness given by the tool.  We have included OH-ness in line 19 to clarify this matter.

  1. Lines 28-32: Citation(s) required

Answer:  A reference is included.

  1. Lines 28-32: Citation(s) required

Answer:  A reference is included.

  1. Lines 35-42: Citations required to validate various categorical statements and claims made

Answer:  References are included

  1. The meaning, concept, relevance and applicability of “OH–EpiCap tool” in AMR surveillance in Norway should be well explained in the introduction

Answer: The meaning, concept, relevance and applicability of “OH–EpiCap tool” in AMR surveillance in Norway could not be discussed in the introduction, if not taking into account of the results of the evaluation beforehand. However, we added some more general aspects of the tool at line 65-69, as we realize that having the methodology section as the last section does not inform the reader enough of the tool before the results and discussion section: Further, “system-specific profiles of existing surveillance interoperability between sectors, highlighting both strength and gaps in surveillance capacity and capabilities are developed by the tool” [10] . The EpiCap tool therefore could be a first step for an internal evaluation of the OH-ness of the AMR surveillance in Norway, which could give guidance for possible improvements.

  1. Figure legends 1-4: remove periods (.) after the last word in the legend.

Answer: Thanks for noticing, it is corrected.

  1. Tables: Where is Table 1? Tables should be cited chronologically i.e. Table 1 before Table 2.  Again, remove the periods (..) after the Table 2 title.

Answer: This has been changed and corrected accordingly.

  1. Lines 96-97: The first evaluation of OH-ness of AMR surveillance in Norway or the first evaluation of OH-ness of AMR surveillance in Norway using the OH-EpiCap tool?

Answer: To our knowledge, this is the first evaluation of the OH-ness of AMR surveillance in Norway, thereby also the first evaluation using the OH-EpiCap tool. We think the text as given, therefore is suitable.

  1. Lines 131-134: This should be mentioned as a limitation of this study:

Answer: We don’t think this is a limitation of the study. However, it can been seen as a limitation of the usefulness of the tool as it does not require all sectors to be included in an AMR surveillance for scoring a 4

  1. Discussion: The OH-ness of the 68% overall AMR surveillance is largely not discussed.  Again, the authors need to discuss the significance of low 57% OH-ness obtained for the operationalization component of the triad and suggest how the enhance operationalization with the OH-EpiCap tool.

Answer:  The overall score of 68% is an average across the three dimensions, and these three dimensions are discussed separately in the manuscript. Together these three are a discussion of the overall 68% score. Text added line 136-137, as well as at line 168-169 we added lines to explain the low score of 57% of the dimension operation, to explain this. 

Round 2

Reviewer 1 Report

The authors describe the use of recently developed OH–EpiCap tool and previously published (Front. Vet. Sci., 24 March 2023 Sec. Veterinary Epidemiology and Economics Volume 10 - 2023 | https://doi.org/10.3389/fvets.2023.1107122). According with the methodology, this tool was applied by a group of stakeholders (key persons in the Norwegian AMR surveillance programmes) in a meeting and the evaluation was performed by using a consensus approach. However, there are many issues with regards to the OH-EpiCap tool use. First,  What were the criteria to define the core group of key persons that are involved in the NORM and NORM-VET, according the methodology?, Second, How many surveys applied during the meeting?, really has statistical significance?, Third, the criteria used for the scoring system used in OH-EpiCap are according with the reference 9, but the status of this paper is:

OH-EpiCap: a semi-quantitative tool for the evaluation of One Health epidemiological surveillance capacities and capabilities, Henok Ayalew Tegegne, Carlijn Bogaardt, Lucie Collineau, Géraldine Cazeau, Renaud Lailler, Johana Reinhardt, Emma L. Taylor, Joaquin Prada, Viviane Hénaux

doi: https://doi.org/10.1101/2023.01.04.23284159

This article is a preprint and has not been peer-reviewed [what does this mean?]. It reports new medical research that has yet to be evaluated and so should not be used to guide clinical practice.

Therefore, the criteria need to be more carefully evaluated with statistical significance for an adequate  Norwegian AMR surveillance systems.

Fourth, from my point of view, since it does not fulfil the criteria set out above, therefore it cannot be approved to be published in Antibiotics.

Author Response

Reply to Reviewer 1:

The authors describe the use of recently developed OH–EpiCap tool and previously published (Front. Vet. Sci., 24 March 2023 Sec. Veterinary Epidemiology and Economics Volume 10 - 2023 | https://doi.org/10.3389/fvets.2023.1107122). According with the methodology, this tool was applied by a group of stakeholders (key persons in the Norwegian AMR surveillance programmes) in a meeting and the evaluation was performed by using a consensus approach. However, there are many issues with regards to the OH-EpiCap tool use.

First, What were the criteria to define the core group of key persons that are involved in the NORM and NORM-VET, according the methodology?,

  • Answer: We have added text to better explain this at line 339-344. According to the OH-EpiCap methodology, which can be found in the OH-EpiCap user guide at page 1,”The questionnaire should be completed by a panel of representatives from the different sectors across the entire surveillance chain, during a workshop.”  It further recommends a panel of 8-10 participants. Although the current study only included five participants in the panel, the participants involved has a long experience with the surveillance of AMR in Norway, covering everything from design and management of the programmes, laboratory methods, data analysis and data curation, dissemination and communication with relevant stakeholders including the government, industry, hospitals and the public. Although, there is a lack of involvement from the environmental sector in the panel per se, only mapping surveys have officially been performed in the environmental sector and several of the participants have been involved in these surveys.

Second, How many surveys applied during the meeting?, really has statistical significance?,

  • Answer: We applied the tool as described in a joint meeting so it is only one survey with in total 48 questions now a bit more described in the material and methods section. Such a survey/ study like this is not performed to have any statistical significance as the evaluation should be regarded as a process and the scores were agreed on throughout the meeting.

Third, the criteria used for the scoring system used in OH-EpiCap are according with the reference 9, but the status of this paper is: OH-EpiCap: a semi-quantitative tool for the evaluation of One Health epidemiological surveillance capacities and capabilities, Henok Ayalew Tegegne, Carlijn Bogaardt, Lucie Collineau, Géraldine Cazeau, Renaud Lailler, Johana Reinhardt, Emma L. Taylor, Joaquin Prada, Viviane Hénaux doi: https://doi.org/10.1101/2023.01.04.23284159

This article is a preprint and has not been peer-reviewed [what does this mean?]. It reports new medical research that has yet to be evaluated and so should not be used to guide clinical practice.

  • Answer: This paper is under review but available as a preprint and we have been kindly asked by the authors/ developers of the tool to refer to this paper.

Therefore, the criteria need to be more carefully evaluated with statistical significance for an adequate  Norwegian AMR surveillance systems.

  • Answer: One purpose of the present study was to use the OH-EpiCap tool to perform the evaluation. The tool do not include the use of statistical significance. To include such statistical tools, a totally different evaluation tool would have to been used, and this was out of scope for the current study. 

Fourth, from my point of view, since it does not fulfil the criteria set out above, therefore it cannot be approved to be published in Antibiotics.

  • Answer: In our view, the paper fulfils the aim of the journal. Furthermore, we have done our best to improve the paper in accordance with the comments of both the editor and the reviewers, and replied and explained in relation to the criteria above.